# Role of miRNAs in Human T Cell Leukemia Virus Type 1 Induced T Cell Leukemia: A Literature Review and Bioinformatics Approach

**DOI:** 10.3390/ijms23105486

**Published:** 2022-05-14

**Authors:** Caio Bezerra Machado, Leidivan Sousa da Cunha, Jersey Heitor da Silva Maués, Flávia Melo Cunha de Pinho Pessoa, Marcelo Braga de Oliveira, Rodrigo Monteiro Ribeiro, Germison Silva Lopes, Manoel Odorico de Moraes Filho, Maria Elisabete Amaral de Moraes, André Salim Khayat, Caroline Aquino Moreira-Nunes

**Affiliations:** 1Department of Medicine, Pharmacogenetics Laboratory, Drug Research and Development Center (NPDM), Federal University of Ceará, Fortaleza 60430-275, CE, Brazil; caio.bmachado97@gmail.com (C.B.M.); flaviamelop@outlook.com (F.M.C.d.P.P.); odorico@ufc.br (M.O.d.M.F.); betemora@ufc.br (M.E.A.d.M.); 2Unichristus University Center, Faculty of Biomedicine, Fortaleza 60430-275, CE, Brazil; leidivansc@gmail.com; 3Hematology and Transfusion Medicine Center, University of Campinas, Campinas 13083-970, SP, Brazil; jerseymaues@gmail.com; 4Department of Biological Sciences, Oncology Research Center, Federal University of Pará, Belém 66073-005, PA, Brazil; oliveira.mb23@gmail.com (M.B.d.O.); khayatas@gmail.com (A.S.K.); 5Department of Hematology, Fortaleza General Hospital (HGF), Fortaleza 60150-160, CE, Brazil; rmonteiroribeiro@icloud.com; 6Department of Hematology, César Cals General Hospital, Fortaleza 60015-152, CE, Brazil; germison@gmail.com; 7Department of Health Sciences, Northeast Biotechnology Network (RENORBIO), Itaperi Campus, Ceará State University, Fortaleza 60740-903, CE, Brazil

**Keywords:** HTLV-1, T cell leukemia, miRNAs, carcinogenesis

## Abstract

Human T cell leukemia virus type 1 (HTLV-1) was identified as the first pathogenic human retrovirus and is estimated to infect 5 to 10 million individuals worldwide. Unlike other retroviruses, there is no effective therapy to prevent the onset of the most alarming diseases caused by HTLV-1, and the more severe cases manifest as the malignant phenotype of adult T cell leukemia (ATL). MicroRNA (miRNA) dysfunction is a common feature of leukemogenesis, and it is no different in ATL cases. Therefore, we sought to analyze studies that reported deregulated miRNA expression in HTLV-1 infected cells and patients’ samples to understand how this deregulation could induce malignancy. Through in silico analysis, we identified 12 miRNAs that stood out in the prediction of targets, and we performed functional annotation of the genes linked to these 12 miRNAs that appeared to have a major biological interaction. A total of 90 genes were enriched in 14 KEGG pathways with significant values, including TP53, WNT, MAPK, TGF-β, and Ras signaling pathways. These miRNAs and gene interactions are discussed in further detail for elucidation of how they may act as probable drivers for ATL onset, and while our data provide solid starting points for comprehension of miRNAs’ roles in HTLV-1 infection, continuous effort in oncologic research is still needed to improve our understanding of HTLV-1 induced leukemia.

## 1. Introduction

In 1979, human T cell leukemia virus type 1 (HTLV-1) was identified as the first pathogenic human retrovirus [1]. It is the cause of adult T cell leukemia (ATL) and is distributed worldwide, although endemic in certain regions of the world, making it a serious global health problem [2,3]. The Southwestern part of Japan, sub-Saharan Africa and South America (in particular Brazil, Colombia, Chile, and Peru), the Caribbean area, Australo-Melanesia, and the Middle East are considered endemic regions, but a study carried out by Gessain and Cassar in 2012 estimated that there are 5 to 10 million individuals infected with HTLV-1 alone. This is, however, a superficial estimation, and the number may be much higher [3].

Brazil is possibly the country with the highest absolute number of HTLV cases in the world, although Japan is the country with the highest prevalence of HTLV-1 infection. Estimates from the Brazilian Ministry of Health indicate that between 700,000 and 2 million people are infected, and although the virus has been among the infectious agents in the country’s blood centers since 1993, there is no specific national policy for HTLV. Most carriers are asymptomatic, and there are more case reports of HTLV-2 infections in people who inject drugs and in the indigenous population [4,5,6,7].

Unlike other retroviruses, such as the Human Immunodeficiency Virus (HIV), there is no effective therapy to prevent the onset of the most alarming diseases caused by HTLV-1 [8]. Outcomes range from hypersensitivity reactions such as arthritis and uveitis to the fatal inflammatory neurologic disorder that is HTLV-1 associated myelopathy/tropical spastic paraparesis (HAM/TSP), HTLV-1 Associated Infectious Dermatitis (IDH), and the malignant phenotype of ATL, an aggressive lymphoproliferative disease which was first described in Japan in 1977 [9].

### 1.1. HTLV-1 Viral Structure and Infection Mechanisms

HTLV-1 is the causative agent of ATL, a malignant tumor of CD4^+^ T cells. It is a retrovirus of the deltaretrovirus genus, and several subtypes of HTLV were discovered later—HTLV-2, HTLV-3, and HTLV-4. It has a structural organization like all retroviruses; two mature viral glycoproteins derived from a common precursor, an outer surface protein (SU) associated with a transmembrane (TM) protein, which is responsible for anchoring the SU-TM complex on the surface of the infected cell, an envelope consisting of a lipid bilayer of cellular origin, and the matrix protein located next to the membrane [10,11,12,13].

It is an enveloped double-stranded RNA virus, and its genome contains three structural genes: *gag*, *pol,* and *env,* and two regulatory genes, *tax* and *rex*. The *tax* and *rex* genes regulate the transactivation of viral replication, in addition to regulating the expression of viral proteins and the basic leucine zipper factor (HBZ) of HTLV-1, which are essential for the maintenance of viral persistence and pathogenesis, in addition to helping in the oncogenic process of ATL, stimulation of tumor growth and development [12,13,14,15].

Viral integration into the host’s DNA happens preferably at transcriptionally active regions, specifically near transcription factor binding sites. HTLV-1 uses different strategies to induce neoplastic transformation, but the immunogenic profile of the host is associated with inflammatory responses, promoting or protecting against the development of HAM or ATL. Its replication occurs parallel to the cell during mitosis, and its infection can occur vertically, during childbirth or breastfeeding, or parenterally, which consists of transfusions, transplants, intravenous drug use, and unprotected sexual intercourse [3,9,15,16,17,18].

Furthermore, HTLV infects mainly T lymphocytes but is also capable of infecting monocytes and dendritic cells due to its direct interaction with glucose transporter 1 (GLUT 1), which is ubiquitously expressed on cell surfaces (Figure 1) [18,19,20,21].

After infection and entry into the cell, viral RNA is reverse transcribed into viral DNA or provirus. After that, the virus can integrate into the host cell genome and exploit normal cell physiology. Viral integrase recognizes and binds to viral long terminal repeats (LTRs) in viral DNA, forming a pre-integration complex (PIC). PIC associates with the host enzyme, protein phosphatase 2A, and engages with host DNA in regions of open chromatin. In addition, HTLV-1 increases genomic instability by direct actions on the DNA, and, in turn, genomic instability can also alter the proviral genome that is often mutated or deleted, thus creating an escape from the host’s immune system [22,23,24,25,26].

During infection, there is viral transmission by cell–cell contact due to an organizing center of polarized microtubules at the cell–cell junction and a virological synapse triggered by Tax protein. From there, the HTLV-1 Gag complex, viral RNAs, and enveloped HTLV-1 virions accumulate at the synapse and migrate into the uninfected cell [27,28,29]. Another way in which viral transmission happens is by using the cell itself through proliferation. That is, the virus integrates into the host genome, and after mitosis, it is transmitted to the daughter cell [23,24].

### 1.2. Leukemogenesis Pathways to Adult T Cell Leukemia

First reported in Japan in 1977, ATL is a lymphoproliferative disease that presents as an oligoclonal or monoclonal expansion of HTLV-1-infected T cells that occurs decades after infection. ATL onset is multifactorial, involving factors related to the virus and the host’s immune and inflammatory responses [10,15].

The disease is classified according to clinical characteristics into acute, chronic, lymphoma-like, and smoldering. Its manifestations include malaise, fever, weight loss, jaundice, skin lesions, adenomegaly, hypercalcemia, elevation of lactate dehydrogenase, and clinical alterations in the number of leukocytes in peripheral blood, with the presence of atypical lymphocytes [10,30,31].

In the acute phase, it is also possible to find lymphocytes with peculiar characteristics called floral cells in the peripheral blood smear, which is a characteristic clinical finding of ATL development. The diagnosis for HTLV-1 infection is performed by ELISA and confirmed by PCR and Western blot, and a worse prognosis may be expected in the acute and lymphoma forms, being more aggressive and with a median survival of one year [24,27,32].

ATL cells’ clonal expansion is promoted by the accessory proteins of HTLV-1 and the Tax protein. Tax utilizes several mechanisms for cellular transformation, including the creation of chromosomal instability, amplification of centrosomes, abrogation of DNA repair, activation of cyclin-dependent kinases and nuclear factor-kB (NF-kB), and AKT signaling, and even silencing of TP53 checkpoints. The maintenance of the ATL transformation occurs through HBZ protein activity, in addition to this protein being essential for the establishment of persistent viral infection [27,28,33,34].

### 1.3. MicroRNAs

The molecular mechanisms of HTLV-1-mediated transformation and carcinogenesis are still unknown; however, there are studies that correlate HTLV carcinogenesis to microRNAs (miRNAs), which are endogenous 18–25 nucleotide long RNAs that play regulatory roles in cell metabolism, directing messenger RNAs (mRNA) for cleavage or translational repression [35,36]. They are involved in the maintenance of a variety of biological processes, including the cell cycle, post-transcriptional gene expression, and other processes [37,38].

MiRNAs genes can be located within the introns and exons of protein-coding genes or in intergenic regions and are transcribed by RNA Polymerase II and processed by the Drosha and Dicer enzymes [38,39,40]. Deregulation of these miRNAs is a common feature in human tumors and has been described in several types of human cancer, such as lung cancer, colorectal cancer, pancreatic endocrine, and even chronic lymphocytic leukemia [41].

Discovering miRNAs, identifying their targets, and clarifying their functions has been a critical strategy for understanding normal biological processes and their roles in disease development [42]. Furthermore, the miRNA expression pattern can be correlated with cancer, so the miRNA profile can be used as a tool for cancer diagnosis and prognosis [43]. Therefore, several studies demonstrate that the cellular expression of miRNAs is affected in HTLV-1 infected cells [44,45].

For this reason, in this study, we sought to identify in previously published works which miRNAs were altered in ATL; in addition, we performed bioinformatics analysis in order to describe possible pathways of malignant transformation and biological interactions to predict the role of these miRNAs in HTLV-1-mediated leukemogenesis. 

## 2. Literature Review and Bioinformatic Analysis Approach

The results are presented below in Table 1, describing the studies that demonstrated miRNAs’ altered expression after HTLV-1 infection in biological samples. After the literature review process, we used the data generated by the 12 published works presented in Table 1 to investigate the role of those miRNAs in the pathogenic model of ATL induced by HTLV-1.

In this study, a list of 42 miRNAs, pointed out in the studies above, was input to the miRWalk algorithm (version 3.0), and for prediction of individual target genes using TargetScan, (version 7.2), miRDB (version 6.0), and miRTarBase (version 8.0), generating three unique files for further analysis [59,60,61]. TargetScan is a predictor that generates predicted interactions, while miRTarBase and miRDB provide validated interactions. A (*p* > 0.95) for binding probability [62], preferentially conserved sites within the 3′ UTR (untranslated region) were applied to the data filtering, and preferably all validated interactions were maintained [63].

After that, we simulated miRNA-gene interaction networks that were performed with Cytoscape (version 3.9.1). The data generated by the prediction of the three files obtained with miRWalk were used to construct the three respective miRNA-gene interaction networks. Then, the three networks were merged according to Cytoscape’s manual merge procedure with an intersection operator to obtain a single miRNA-gene interaction network [64].

Lastly, we performed functional annotation of the genes linked to 12 miRNAs that showed a significant biological interaction between all described in the studies, using DAVID (Database for Annotation, Visualization, and Integrated Discovery) [65] to identify the pathways of KEGG (Kyoto Encyclopedia of Genes and Genomes) [66]. All target genes linked to miRNAs (miR-34a-5p, miR-146b-5p, miR-181b-5p, miR-26a-5p, miR-26b-5p, miR-222-3p, miR-155-5p, miR-193a-5p, miR-199a-3p, miR-199b-3p, miR-423-5p, miR-150-5p) were investigated with DAVID using the Refseq identifier of the species (homo sapiens). A total of 90 genes were enriched in 14 KEGG pathways with significant values annotated with (−Log10 *p*-value) < 0.05. An interactive network was built with Cytoscape to show all target genes enriched in KEGG pathways. The topologies of all networks were constructed with network density < 0.05.

Most miRNAs described in these studies were mentioned only once, and a correlation between authors and cited miRNAs can be found in Appendix A. Almost all authors predicted or hypothesized oncogenic-related cellular pathways that might be deregulated due to miRNA erroneous expression [46,47,48,49,50,51,52,53,54,55,56,57,58]; however, our own bioinformatics approach was able to predict pathways through which these miRNAs described previously might induce carcinogenesis.

The results of the bioinformatics analysis are shown in the figure below (Figure 2). The target predictions were integrated through predictive interactions between miRNAs and their target genes using the Cytoscape tool that generated networks of topologies of significant and representative densities [64]. Using 12 miRNAs that stood out in the prediction of targets, it was possible to predict their interactions using miRNA-Target interaction networks.

There was a significant enrichment of miRNA target genes in important KEGG pathways for cancer, such as hsa05200: Pathways in cancer; hsa04310: WNT signaling pathway; hsa04010: MAPK signaling pathway; hsa04350: TGF-β signaling pathway; hsa04014: Ras signaling pathway, all shown at central interaction network in Figure 2, which presented 39 nodes and 80 edges and a density equal to 0.054. A full scope of all identified pathways and targeted gene ontology can be found in Appendix A.

While the deregulation of any signaling pathways is sure to disrupt cell metabolic homeostasis, some are more relevant for leukemogenesis than others and will be discussed further in this paper. Of importance to note is the presence of *TP53* as a trending gene, appearing as part of most of the identified significantly deregulated pathways.

## 3. *TP53* Mutations and Modulation in HTLV-1 Infection

The loss or mutation of the *TP53* tumor suppressor gene is a very common genetic lesion in human cancer, so much so that the inheritance of a germline mutation in *TP53* generates increased risks of developing neoplasms. The reason for this well-reported carcinogenesis is the tumor suppression functions performed by the protein that is commonly called the guardian of the genome, as it protects DNA integrity [67,68,69].

According to Borrero and El-Deiry [68], the *TP53* gene is mutated in about 50% of human cancers, but there is also a biological inactivation of its pathway in the others. It can be altered directly or by targeting the oncoproteins MDM4 regulator of p53 (MDM4) and MDM2 proto-oncogene (MDM2) that regulate *TP53* activity, thus being altered indirectly.

In response to stress factors such as DNA damage, hypoxia, UV irradiation, oxidative free radicals, oncogenes, or growth factor deficiencies, the TP53 protein is activated and can modulate target genes that regulate cell cycle arrest, apoptosis, DNA repair, angiogenesis, and metastasis [68,69,70,71].

*TP53* importance in carcinogenesis is a well-established concept, and several different miRNAs have been reported to silence *TP53* expression directly or indirectly, while *TP53* also regulates miRNAs expression, making it clear that this regulation can be associated with malignant changes [68,70]. In ATL cases, inhibition of TP53 activity is associated with HTLV-1 infection due to Tax and HBZ, thus promoting its malignant proliferation [72,73].

Studies by Nascimento et al. [46] and Ruggero et al. [51] both report deregulation of miR-150-5p, which by our analysis demonstrated a direct interaction with *TP53* activity. There is, however, controversy when determining miRNA expression levels in HTLV-1 infection, with the former reporting an upregulated expression of miR-150-5p levels in HTLV-1 asymptomatic carriers (ACs) when compared to healthy controls (HC) and ATL patients, and the latter reporting a downregulation of miR-150-5p in HTLV-1 infected cells when compared to HC.

It is important to note that, while the downregulation of a miRNA that interferes with TP53 activity may seem beneficial for proper cell cycle control, the modulation of TP53 and ATL onset may also be correlated with *TP53* mutational profile. *TP53* status as a tumor suppressor may be easily overridden even by single-nucleotide mutations that are able to induce gain of oncogenic functions (GOF) for TP53 protein and dominant-negative effects (DNE) over wild-type *TP53*. Oversimplifying the past statements, downregulating *TP53* may actually be beneficial for the proper regulation of cell homeostasis if its mutations are inducing an oncogenic-protein phenotype and disrupting its physiological regulating activities [74,75].

This difference in miRNA expression levels may be representative of proliferation and survival advantages that cells obtain in in vivo models during disease development that are not seen when compared to HTLV-1 infected cell lines in vitro. Nascimento et al. [46] described that ATL patients had increased levels of miR-150-5p compared to HC but lower levels when compared to HTLV-1 ACs; this may be what differentiates asymptomatic infections from ATL onset. It is possible to hypothesize that genomic instability driven by HTLV-1 integration into the host’s genome leads to a series of mutations in genes critical for maintaining cell metabolism and, in the case of *TP53*, overexpression of miRNAs modulating genetic GOFs and DNEs is what keeps leukemogenesis in check [25,26,46].

Furthermore, the reported downregulation of miR-150-5p in HTLV-1 infected cell lines compared to samples of HC by Ruggero et al. [51] only corroborate the idea that TP53 GOFs and DNEs are the main effectors for survival advantage gains in HTLV-1 infections and consequent achievement of malignant phenotypes. Regardless of miRNA levels, though, *TP53* mutations and rearrangements are important prognostic factors and major predictors of response to treatment in ATL patients [76,77].

## 4. Pathways Identified as Deregulated in ATL

Beyond *TP53*, miR-150-5-p validated targets include other proto-oncogenes, such as *c-Myb* and the *NOTCH3* gene [51]. The *c-Myb* gene, which encodes transcription factors that regulate cell proliferation, differentiation, and apoptosis, is found to be overexpressed in a wide range of human cancers, and leukemia studies have associated its expression with a worse prognosis and its transcript physically binds to TP53 [78,79].

Another validated target of miR-150-5p is the *NOTCH3* pathway, which regulates cell proliferation, differentiation, cell fate determination, and stem/progenitor cell self-renewal in adult and embryonic organs. Elevated expression is increasingly documented in cancers and is correlated with rapid malignant progression, abnormal differentiation, metastasis, and worse prognosis. There is also a known interaction of NOTCH signaling with other major signaling pathways such as WNT and TGF-β [80,81].

### 4.1. WNT Canonical and Non-Canonical Pathways

The WNT signaling pathway is associated with cell differentiation, polarization, and migration during development and is fundamental in cell growth control. Its main component is the family of WNT proteins that activate cell membrane receptors in a paracrine and autocrine manner. It is based on three signaling pathways: WNT canonical signaling pathway (CP), which is β-catenin-dependent, and two WNT non-CP that are β-catenin-independent: planar cell polarity (PCP) signaling pathway and a WNT/Ca^2+^ pathway that has Wnt5a as the main effector. This pathway mainly plays a pro-tumor role, and WNT/β-catenin signaling is activated in many types of cancer. Its primary receptors are frizzleds (FZD), which are transmembrane G-coupled proteins. WNT signaling is inhibited by endogenous inhibitors and secreted FZD-related proteins (sFRPs) that interact directly with WNTs [82,83,84].

However, Ma et al. [85] previously reported that ATL cells rely on the WNT non-CP of WNT/Ca^2+^ for tumorigenesis development while inhibiting the WNT CP, which seems to have inhibitory effects on ATL cell growth through HBZ mediated activity. The miRNA expression profiles identified in our analysis support these observations and corroborate an excessive signaling of Wnt5a.

An overactivation of Wnt5a may be observed in ATL cells mediated by increased expression of c-myb, which is expected to happen at ATL onset after downregulation of miR-150-5p expression [46,51,86]. Also, phospholipase C beta (PLCB) are a family of direct downstream effectors of Wnt5a, and we identified miR-423-5p, which was reported as downregulated in HTLV-1 infected patients by Fayyad–Kazan et al. [47], to directly interact with PLCB1 expression [87]. Therefore, it is possible that HTLV-1 induces WNT non-CP both through increased activation of Wnt5a but also through upregulation of its downstream signaling effectors.

Other miRNAs that we identified as interacting with the WNT pathway, however, are much more associated with HTLV-1 capability of inhibiting the CP. It is worth noting the upregulation of miR-34a-5p in HTLV-1 infected cell lines since we were able to identify its interaction with the transcription factor lymphoid enhancer-binding factor 1 (LEF1), which is a main effector of WNT CP and was already reported to be suppressed by HBZ in HTLV-1 infection [51,85]. In Table 1 also, downregulation of miR-222-3p and miR-146b-5p was associated with ALT onset, and both miRNAs interact directly with inhibitors of WNT CP, those being TLE family member 3 (TLE3) and zinc and ring finger protein 3 (ZNRF3), respectively [46,51,88,89].

This particular profile of respective suppression and overactivation of canonical and non-canonical WNT pathways is not unexpected, even when disregarding miRNA activity since both pathways are activated by different ligands that interact in a reciprocal inhibitory fashion due to their competitive binding with the same FZD membrane receptors [90,91].

### 4.2. TGF-β Dual Roles in Cancer

Transforming growth factor-β (TGF-β) signaling is essential during embryo development and associated with many important biological processes; however, TGF-β roles in cancer are diverse and dependent on other cell signaling pathways, being considered a tumor suppressor at early tumor stages and an enhancer of malignant phenotypes later in cancer development [92,93,94].

TGF-β achieves its role as a tumor suppressor due to the ability to regulate cell proliferation, differentiation, and immune cell modulation through the activity of its main signal transducers, the SMAD family member (SMAD) proteins. Downstream of TGF-β activation, SMADs interact with cyclin-dependent kinase inhibitors (CDKNs), inhibitors of DNA binding (IDs) proteins and other proliferation drivers to modulate genomic expression, cell cycle, and apoptosis induction [95,96,97].

The oncogenic role of TGF-β, however, is highlighted at later stages of tumor development, where overactivation of TGF-β signaling is associated with increased potential of invasion and metastasis, as well as with downregulation of immune cell activity in the tumor microenvironment [97,98].

SMAD2, zinc finger FYVE-type containing 16 (ZFYVE16), and Sp1 transcription factor (SP1) are signal transducers acting downstream of TGF-β that we identified to be regulated by the activity of miR-155-5p, miR-222-3p, and miR-150-5p, respectively. All of the three previously mentioned miRNAs were reported by Nascimento et al. [46] as upregulated at HTLV-1 infection but downregulated at ATL onset, and this probably points towards the expected panorama of TGF-β inhibition at early stages followed by later overexpression [98].

ZFYVE16 is a direct enhancer of SMAD2 activity, as well as other SMAD members, and was reported by Zhao et al. [99] as an inhibitor of T cell proliferation through modulation of TGF-β signaling. In addition, SP1 is a transcription factor responsible for modulating the activity of a myriad of genes related to oncogenesis and is also an effector for cell cycle arrest through CDKN2B-mediated activity [100,101]. Their inhibition by miRNAs upregulation could mean a proliferation advantage for T cells mediated by HTLV-1 at initial infection.

It is also of note that, after HTLV-1 integration in the host genome, the integrated viral DNA harbors multiple binding sites for SP1 at viral gene promoters. While the roles of all of these binding sites as repressors or enhancers of gene expression are not fully elucidated, SP1 activity seems to be an important regulator for latent HTLV-1 activation in response to cell stressing factors [102,103].

Beyond the above-mentioned interactions, TGF-β inducing factor homologous box 2 (TGIF2) and E2F transcription factor 5 (E2F5), both of which act downstream of TGF-β signaling, have also had an identified interaction in our analyses. Mir-34a-5p, reported by Ruggero et al. [51] as upregulated in HTLV-1 infected cells, acts directly over the expression of both proteins. The transcription factor TGIF2 belongs to the TALE homologous domain protein family, including TGIF, TGIF2, and TGIF2LX/Y, while E2F5 is a transcription factor belonging to the E2F family, which is composed of eight members ranging from E2F1 to E2F8 [104,105].

TGIFs are known antagonists of TGF-β signaling due to their repressor effects over SMAD2/3 activity. By inhibiting the formation of the E1A binding protein p300 (p300)/SP1 transcriptional complex, which acts downstream of SMAD2/3 activity, TGIF2 negatively regulates p300/SP1-mediated cell cycle arrest and thus is implied as a cofactor for carcinogenesis in a variety of tumors [106,107,108,109]. On the other hand, E2F5 acts alongside transcription factor Dp-1 (DP1) and RB transcriptional corepressor like 1 (p107) to repress the activity of the proliferation driver c-myc in a SMAD2/3 dependent way [110,111,112]. Downregulation of both genes by the increased activity of miR-34a-5p in ATL cells possibly defines a profile for overactivation of TGF-β signaling by HTLV-1 during ATL development.

Furthermore, Zhao et al. [113] described TGF-β signaling as essential for successful HTLV-1 persistence of infection as HBZ-mediated activity causes the overactivation of SMAD2/3 response and increased formation of p300 transcriptional complex, leading to modulation of T cell phenotypes and escape from the host’s immune system.

### 4.3. RAS and MAPK Signaling

Another important interaction of miR-34a-5p was with RAS and MAPK pathways, through *RAS-related* (*RRAS*) gene and platelet-derived growth factor receptor (PDGFR), specifically PDGFRA, a classic proto-oncogene encoding receptor tyrosine kinases [114].

Members of the RAS family that contains more than 100 proteins, the RRAS subfamily is composed of only 3 small proteins: RRAS, RRAS2, and MRAS, which are molecular switches that toggle between GTP-bound and GDP-bound conformations. It has been reported that RRAS is involved in several cellular functions, including enhancement of integrin function, regulation of cell adhesion, invasion, and migration, and related to different mechanisms that regulate its oncogenic potential, one of which is epigenetic regulation. The widespread prevalence of RAS mutations in human cancer has been recognized for many years. In cancer, members of the RRAS subfamily have been implicated in anchorage-independent growth, increased invasiveness, stimulation of angiogenesis, and tumorigenicity. In addition, RRASs have been linked to leukemias, and their overexpression can impair tumor-targeted lymphocytes, indirectly promoting tumor growth [115,116,117,118].

MAPK is, in its classical pathway, an extension of RAS signaling through the RAS-RAF-MEK-ERK signaling cascade, which regulates genes that control cell development, differentiation, proliferation, and apoptosis. It plays a central role in human cancer, and many of its elements have been identified as oncogenes, being overactivated in a wide variety of tumors. In addition, aberrant activations of MAPK signaling and its interactions with other cell signaling pathways in the pathogenesis of chronic lymphocytic leukemia have also been described [119,120,121].

Interestingly though, the upregulation of miR-34a-5p reported by Ruggero et al. [51] in HTLV-1 infected cells would possibly point towards a disruption of both of the above-mentioned pathways through inhibition of receptor expression for extracellular growth factors and decreases in RRAS expression and phosphorylation levels. The RAS pathway, however, is activated intracellularly through multiple RAS isoforms, such as KRAS, HRAS, and NRAS, and by many growth and colony-stimulating factors, and the simple inhibition of RRAS isoform and PDGFRA may not be sufficient to inhibit RAS and MAPK activities altogether in ATL leukemogenesis [122,123]. The importance of RAS signaling for ATL cells is even highlighted by the fact that Tax-mediated anti-apoptotic effects are dependent on cell levels of phosphorylated RAS [124,125].

Still adjacent to MAPK signaling is the activity of the tumor necrosis factor (TNF) family and their ambiguous role in tumor development [126,127]. MiR-146b-5p, reported by Ruggero et al. [51] as downregulated in HTLV-1 infected cells, was identified to interact with TNF receptor-associated factor 6 (TRAF6), a transducer of TNF signaling that ultimately leads to increased expression of NF-kB [128]. The role of NF-kB in tumorigenesis is well established, and it is no different in ATL onset, where Tax potential to induce oncogenesis is directly linked to NF-kB at early tumor development, and NF-kB increased activity is even able to be maintained at later stages, regardless of Tax expression profile [129,130,131].

Through a bioinformatic approach, we were able to predict many interactions of miRNAs in important pathways in carcinogenesis that were correlated with HTLV-1 infection, and Figure 3 represents some of these targeted pathways and the miRNAs involved in their regulation.

## 5. Conclusions

Several studies reveal deregulated miRNAs in HTLV-1 infection, and our bioinformatic analysis was able to predict several of these miRNAs’ interactions with important pathways in tumorigenesis, especially TP53, WNT, RRAS, TGF-β, and MAPK, thus being able to address them as probable drivers for ATL onset. While the data generated by our bioinformatics tools are able to provide solid starting points for further comprehension of miRNAs’ roles in HTLV-1 infection, the continuous effort in oncologic research to clarify still to be elucidated molecular pathways is needed to improve our understanding of HTLV-1 induced leukemia.

## Figures and Tables

**Figure 1 ijms-23-05486-f001:**
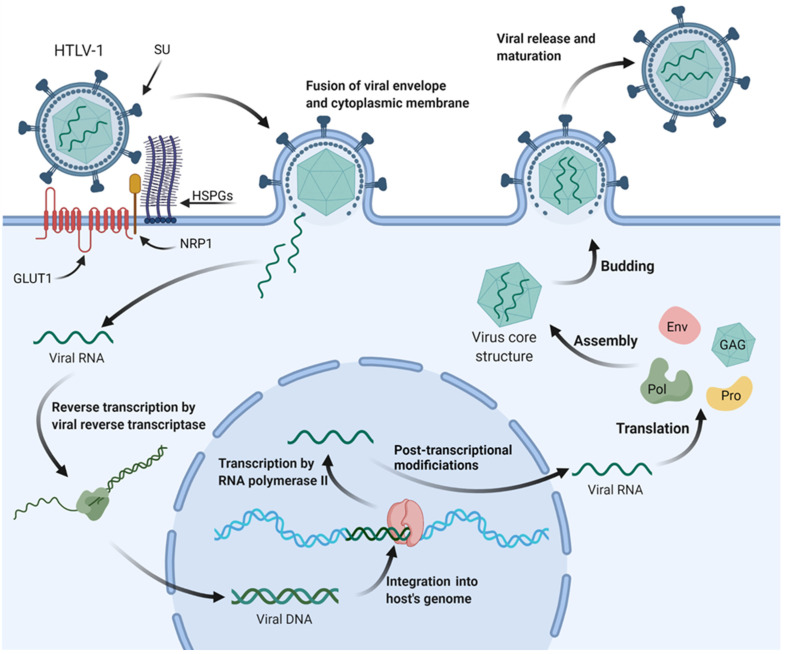
HTLV-1 mechanisms of cell infection and replication. Surface subunit (SU) of HTLV-1 glycoproteins interacts with heparan sulfate proteoglycans (HSPGs) of the targeted cell’s cytoplasmic membrane. A complex is then formed between the viral envelope, HSPGs, neuropilin-1 (NRP1), and glucose transporter-1 (GLUT1). The envelope fuses with the cytoplasmic membrane, and the viral RNA is released on the cytoplasm, where it will be reverse transcribed and carried to the nucleus as viral DNA for integration into the host’s genome. The provirus is then transcribed by RNA polymerase II of the cell’s transcriptional machinery, and after post-transcriptional modifications, the mature viral mRNA is transported back to the cytoplasm. Translation of viral mRNA and alternatively spliced mRNAs generate the proteins necessary for viral assembly inside the host cell, such as envelope glycoprotein (Env), polymerase (Pol), protease (Pro), and structural proteins (GAG). These proteins, alongside two copies of viral RNA, migrate to the budding site and are released from the cell’s surface to further mature into infectious viral particles following protease-dependent activity.

**Figure 2 ijms-23-05486-f002:**
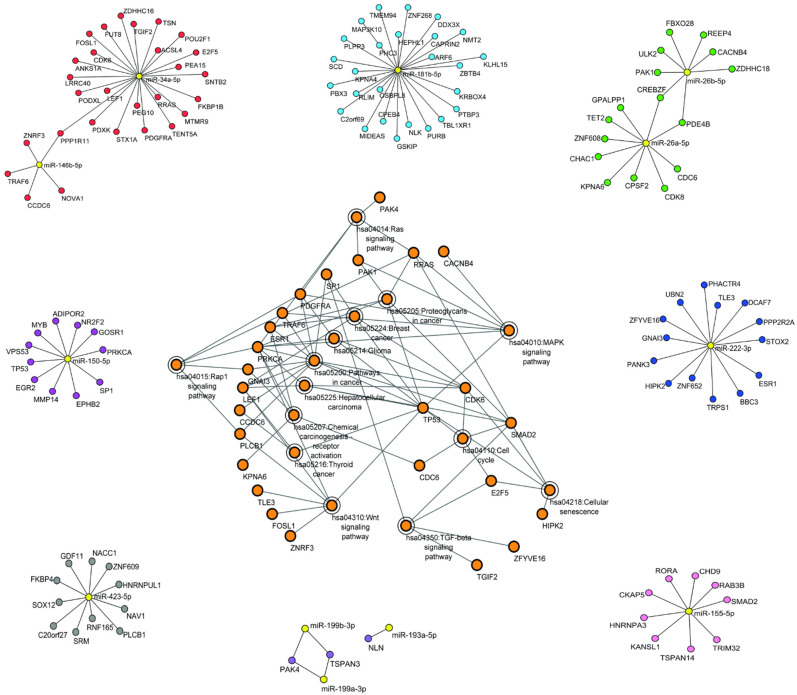
Cytoscape analysis network. A miRNA-gene interaction network was generated from a manual procedure of the intersection of individual networks made from target prediction data generated with miRWalk. It was possible to build all networks of all miRNAs: miR-34a-5p, miR-146b-5p, miR-181b-5p, miR-26a-5p, miR-26b-5p, miR-222-3p, miR-155-5p, miR-193a-5p, miR-199a-3p, miR-199b-3p, miR-423-5p, miR-150-5p. The target genes of these 12 miRNAs were enriched with KEGG and normalized to Log2 (*p*-value). At the center, we propose a wider interaction network with the target genes of these miRNAs with their respective pathways, where many of these pathways are related to cancer.

**Figure 3 ijms-23-05486-f003:**
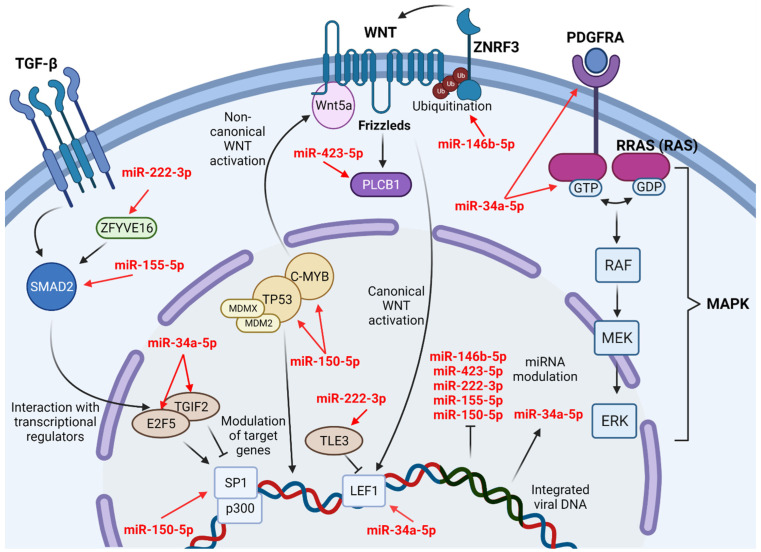
MiRNAs interactions on cellular pathways modulated by HTLV-1 infection driving ATL onset. MiR-34a-5p is an upregulated and extremely relevant miRNA in HTLV-1 cell interactions. Most other miRNAs appear to be downregulated and have a wide range of modulation activity over pathway inhibitors and enhancers. MiR-150-5p interacts directly with TP53 activity and may regulate its activities after oncogenic gains of function. ATL cells also present a preference for WNT non-canonical pathway activation, with downregulation of canonical pathway activity through miRNA modulation and Wnt5a competitive binding. Furthermore, HTLV-1 infected cells also differently express miRNAs involved in the regulation of many other important cell pathways, such as TGF-β, RAS, and MAPK.

**Table 1 ijms-23-05486-t001:** Studies describing altered expression of miRNAs after HTLV-1 infection.

miRNA	Type of Samples Analyzed	Proposed Cellular Pathways	Expression Levels	Reference
miR-199a-3p	Samples of ATLL patients and asymptomatic HTLV-1 carriers	NR	Downregulated when comparing ATL patients to asymptomatic HTLV-1 carriers, but upregulated when comparing ATL patients to healthy controls	[46]
miR-26a-5p		Predicted to target ABHD2, HMGA1, EP400, CDK8, ZNF608, KPNA6, and ZSWIM6		
miR-199b-3p		NR		
miR-150-5p		Predicted to target ADIPOR2, SP1, ZEB1, EGR2, and CBL		
let-7d-3p		NR		
miR-155-5p		Predicted to target MORC3, TRIM32, SMAD2, and TP53INP1		
miR-26b-5p		Predicted to target CREBZF, USP3, KPNA6, and RAP2C		
miR-222-3p		Predicted to target PANK3, TLE3, ZFYVE16, PHACTR4, and SUN2		
miR-181b-5p		Predicted to target ZNF780B, HEPHL1, ZNF268, ZBTB4, PTBP3, NR6A1, PBX3, CAPRIN2, PHC3, C2orf69, INO80D, CPOX, KPNA1, TNPO1, PTEN, GSKIP, ARF6, and MPP5		
miR-30e-3p		NR		
miR-127	Samples obtained from HTLV-1 infected patients at the time of diagnosis	NR	Downregulated in HTLV-1 infected patients	[47]
miR-136				
miR-142-3p				
miR-221				
miR-423-5p				
let-7b			Upregulated in HTLV-1 infected patients	
miR-29c				
miR-30c				
miR-193a-5p				
miR-885-5p				
miR-34a	C91PL, MT-2, HUT-102, C8166, ATL-2, and ED40515(−); Samples from ATLL patients	A transcriptional target of p53, NF-kB, Tap73, and ELK; Targets many cell proliferation and survival pathways such as MYC, MYCN, MET, CCDN1, CDK6, BCL2, and NOTCH1; May modulate expression of tumor suppressor genes	Upregulated both in cell lines, except ED40515(−), and in patient samples when compared to PBMC of healthy donors	[48]
miR-150	MT-4, MT-2, C8166, C91PL, Jurkat, MT-1, ATL-T, ED-40515(−), ALT-25, ATL-43T, LMY1, and ATL-55T; Samples from ATLL patients	Inhibition of STAT1 expression and suppression of STAT1-dependent genes	Downregulated in HTLV-1 infected and ATL-like cell lines	[49]
miR-223				
miR-17	CD4^+^ and CD8^+^ T cells from HTLV-1 infected individuals and healthy donors	Upregulated in an HBZ-dependent manner; Trigger cell proliferation and genomic instability through inhibition of OBFC2A-hSSB2 pathway	Upregulated in CD4^+^ infected clones when compared to uninfected CD4^+^ clones	[50]
miR-21				
miR-23b		Upregulated in an HBZ-dependent manner		
miR-27b				
miR-34a-5p	C91PL and MT-2	Regulator of cell proliferation and survival in a p53-dependent manner; Its upregulation in other virus-associated malignancies suggests diverse cellular effects depending on context	Upregulated in HTLV-1 infected cell lines	[51]
miR-150-5p		Target oncogenes c-Myb and NOTCH-3; Antiproliferative and proapoptotic effects on B-lymphoma, T-ALL, and NK cell lines	Downregulated in HTLV-1 infected cell lines	
miR-146b-5p		Potential activity over TRAF6, IRAK1, FADD, and CXCR4		
miR-155	MT-2, MT-4, C5/MJ, SLB-1, HUT-102, MT-1, and ED-40515(−), Jurkat, MOLT-4, CCRF-CEM, and JPX-9	Upregulation of miR-155 by Tax through activation of NF-kB and AP-1; Potential inhibition of transcriptional repressors BACH1 and HIVEP2	Upregulated in HTLV-1 infected cell lines	[52]
miR-149	Jurkat and MT-2	Act upon histone acetyltransferases p300 and p/CAF, regulating chromatin remodeling	Downregulated in HTLV-1 infected cell lines when compared to Jurkat	[53]
miR-873				
miR-31	Samples of ATLL patients	Regulated by Polycomb proteins activity; Inhibits NF-kB-inducing kinase (NIK)	Downregulated in ATL samples when compared to healthy donors	[54]
miR-146a	MT-2, MT-4, C5/MJ, SLB-1, MT-1, ED-40515(−), HUT-102, Jurkat, MOLT-4, CCRF-CEM, and JPX-9	Upregulation of miR-146a by Tax through activation of NF-kB; Enhances cell growth through undetermined mechanisms; Able to target TRAF6 and IRAK1	Upregulated in HTLV-1 infected cells when compared to non-infected cell lines	[55]
miR-181a	C8166, MT-2, MT-4, HUT102, LAF, MUO4; Samples from ATLL patients	Favors B cell differentiation and regulates T cell receptor signaling	Downregulated in HTLV-1 infected cell lines and ATL patient samples	[56]
miR-132		Involved in innate immunity		
miR-125a		Involved in innate immunity and regulation of regulatory T cells functions		
miR-155		Upregulated through NF-kB and JNK pathways; Regulates dendritic and T cell interactions as well as T helper cells differentiation	Upregulated in HTLV-1 infected cell lines and ATL patient samples	
miR-142-3p		Induces differentiation towards T cell lymphopoiesis		
miR-150		Regulates differentiation of B and T cell lineages	Downregulated in HTLV-1 infected cell lines, but upregulated in ATL patient samples	
miR-223		Induces differentiation towards T cell lymphopoiesis		
miR-142-5p		Induces differentiation towards T cell lymphopoiesis	Upregulated ATL patient samples	
miR-146b		Involved in innate immunity	Downregulated ATL patient samples	
miR-223	Jurkat, HuT-78, CEM, HuT-102, StEd, ATL-3, PaBe, JuanaW, Champ, C91-PL, MT-2, Abgho, Nilu, Eva, Xpos, and Tesi	NR	Downregulated in HTLV-1 infected and ATL-derived cell lines	[57]
miR-21		Predicted binding sites to a cohort of regulatory genes through in silico analysis	Upregulated in HTLV-1 infected and ATL-derived cell lines	
miR-24				
miR-155				
miR-146a		Upregulation of miR-146a by Tax through activation of NF-kB; Predicted binding sites to a cohort of regulatory genes through in silico analysis		
miR-93	MT-1. ATL55T, ATL-2, ATL48T, TLOM1, ED, 43T, MT-4; Samples from ATLL patients	Inhibition of tumor suppressor TP53INP1	Upregulated in HTLV-1 infected cell lines and ATL patient samples	[58]
miR-130b				

miRNA: microRNA; HTLV-1: Human T cell leukemia virus type 1; ATL: Adult T cell leukemia/lymphoma; NR: Not reported; PBMC: Peripheral blood mononuclear cells.

## Data Availability

Not applicable.

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
