# Peer review of "Role of miRNAs in Human T Cell Leukemia Virus Type 1 Induced T Cell Leukemia: A Literature Review and Bioinformatics Approach"

_ijms, 2022, doi:10.3390/ijms23105486_

Round 1

Reviewer 1 Report

In the manuscript entitled “Role of miRNAs in human adult T-cell leukemia virus-1 induced T-cell Leukemia: a literature review and bioinformatics approach” by da Cunha et al., the authors provided a comprehensive review of the recent research progress regarding miRNAs in leukemogenesis of ATL. The authors systematically demonstrated any experimental evidence for readers to understand roles of miRNAs and their relationship in several signaling pathways with their own in silico analysis. There are several comments below.

Major comments:

(1) In Section 3. P53, WNT, RRAS and TGF-beta roles in Carcinogenesis, the authors described general phenomenon in several signaling pathways; however, they did not include any association with tumorigenesis of ATL and the cited experimental evidence. This Section 3 should be discussed regarding miRNA involvement and their roles of the signaling pathways in ATL cells, as summarized in Table 1. Please add relationship between ATL tumorigenesis and miRNA-mediated activation/attenuation of molecules in each signaling pathway.

(2) Refs. 10 and 42 are identical; check the reference lists carefully, and cite appropriate references through the text. For example, line 64, ref. 11 would not be appropriate.

Minor comments:

(1) Delete “adult” from all of human adult T-cell leukemia virus-1” in the Title, Abstract, and Introduction. Also, human T-cell leukemia virus-1 should be corrected as human T-cell leukemia virus type 1.

(2) Lines 60-64 can be moved into 1.4 MicroRNAs.

(3) Line 66, the reviewer could not find any description regarding “Tis virus” in ref.11. Cite proper reference if the historical thing is correct.

(4) Line 87, HTLV infects different cells (Figure 1); does Figure 1 show any mechanisms of HTLV infection in different cells? The title of Figure 1 is HTLV-1 mechanisms of cell infection and replication, thus, “HTLV infects different cells (Figure 1)” did not fit in with the content of Figure 1.

(5) Lines 87-88, can fibroblasts be infected with HTLV-1? Show any original papers describing such a character, or it should be deleted.

(6) Lines 89-90, The diagnosis ---; this sentence can be moved to line 128. Also, lines 136-140 can be moved to line 128 in the same paragraph.

(7) Lines 101, 130, and so on, Tax but not TAX is generally common.

(8) Line 122, What is multifactorial?

(9) Lines 124-125, ATL is basically classified as acute, chronic, smoldering, and lymphoma.

(10) Line 132, nuclear factor kB is generally abbreviated as NF-kB.

(11) In Table 1, the title is Studies describing altered expression of miRNAs after HTLV-1 infection; however, it includes data from other lymphoma and leukemia cell lines, such as HUT-78, Jurkat, MOLT-4, and CCRF-CEM. They should be removed. Further, data from Abgho, Nilu, Eva, and Xpos (from ref. 47) may be removed, because they are derived from HAM/TSP, even they are HTLV-I-infected cells; the reviewer believes that the authors in this review focus on leukemogenesis of ATL, not HAM/TSP.

(12) In Figure 2, word size is too small to see, and some of them are very fuzzy. Please modify it.

(13) Lines 250-254, asymptomatic carriers (ACs), who are infected with HTLV but have not developed the disease. Correct ASP.

Author Response

Dear reviewer, my co-authors and I would like to thank you for the suggestions made during this high-quality review and then we present the answers to the questions.

We inform that with the reviews and suggestions, we were able to improve the idea presented by our work and we appreciate the opportunity. We hope this review has left the article suitable for publication in this high-impact journal and respect in the area.

Kind Regards.

Response to reviewer 1

Major Comments:

(1) In Section 3. P53, WNT, RRAS and TGF-beta roles in Carcinogenesis, the authors described general phenomenon in several signaling pathways; however, they did not include any association with tumorigenesis of ATL and the cited experimental evidence. This Section 3 should be discussed regarding miRNA involvement and their roles of the signaling pathways in ATL cells, as summarized in Table 1. Please add relationship between ATL tumorigenesis and miRNA-mediated activation/attenuation of molecules in each signaling pathway.

R: Section 3 went through extensive revisions and was further divided into two topics so that it could be discussed in more detail. Furthermore, Figure 3 was also updated to better convey what is being detailed in the revised text.

(2) Refs. 10 and 42 are identical; check the reference lists carefully and cite appropriate references through the text. For example, line 64, ref. 11 would not be appropriate.

R: The reference list has been properly standardized and update using automatic reference importers.

Minor Comments:

The suggested revisions have been accepted and corrected along the text. Figure 2 quality was also enhanced for a better reading. However, minor comment (11) was the only one in disagreement with the authors perspectives.

(11) In Table 1, the title is Studies describing altered expression of miRNAs after HTLV-1 infection; however, it includes data from other lymphoma and leukemia cell lines, such as HUT-78, Jurkat, MOLT-4, and CCRF-CEM. They should be removed. Further, data from Abgho, Nilu, Eva, and Xpos (from ref. 47) may be removed, because they are derived from HAM/TSP, even they are HTLV-I-infected cells; the reviewer believes that the authors in this review focus on leukemogenesis of ATL, not HAM/TSP.

R: Although the study focus is indeed ATL leukemogenesis, the other cell lines described in table I were mostly used in the reported studies as comparative controls for miRNA levels and as such are of relevance for the determination of miRNA differential expression.

Reviewer 2 Report

The manuscript written by Leidivan Sousa da Cunha et al describes the role of miRNAs in human adult T-cell leukemia virus-1 induced T-cell Leukemia. The authors reviewed the literature and took bioinformatics approach to analyze studies that reported deregulated miRNA expression in HTLV-1 infected cells and patients’ samples to understand how this deregulation could induce malignancy.

The manuscript has been poorly written. There are loads of typos and misspellings, and references are lacking in places.

Major concerns

The generated data by the used bioinformatic tools could be a starting point to obtain preliminary data to further study the role of miRNAs in HTLV-1 infection. However, this does not provide enough data to publish the findings in its current state.

Other comments and suggestions:

  • Line 62: Please correct. miRNAs are not always 22 bp. Give estimate; approximately 17-25 bp
  • Line 80: HTLV-1 does not integrate randomly – check Melamed A. et al PLoS Path, 2013
  • Line 94-95: reference for PP2A associating with HTLV-1 integrase: Maertens GN NAR 2016.
  • Line 96 …: HTLV-1 increases genomic instability by direct actions to the DNA…this is followed by references in Portuguese. Whilst it is great that these references are included, it would be helpful to the reader to explain in a bit more detail what was described in these papers.
  • Line 101-102: HTLV-1 gag complex and viral RNAs accumulate at the synapse and migrate to the uninfected cell: EM data has shown that viral particles can be found in the cleft created by the virological synapse, not simply RNA and Gag. Majorovits E. et al PLoS One 2008.
  • Figure one can show the cell-to-cell transmission or transmission by mitosis, as described at line 99-105.
  • Figure 1: retroviruses do not assemble with intact capsid cores, the cores become mature following protease dependent maturation
  • Explaining the general knowledge about HTLV-1 infection is too long.
  • Line 111: reverse transcribed
  • Line 112: the provirus is then transcribed, not the proviral load
  • Line 120-123 is totally unnecessary
  • Line 120 ATL has not introduced for the first time. It has already been mentioned before.
  • There is a missing subject before is in line 122.
  • Line 126, the closed parentheses should be a comma.
  • Have you already defined the accessory proteins of HTLV-1?
  • I failed to relate the line 136-140 to the whole concept in general and to the previous paragraph particularly. What is the point of floral cells here?
  • line 146 pol III transcription of miRNAs does occur. See the following papers as example

RNA polymerase III transcribes human micro RNAs published 12 NOV 2006. Glon M Borchet et al, Nature structural and Molecular biology.

       miRNAs: Biogenesis, function and role in cancer. published 2010

       Leigh Ann mac farlane et al. Curr Genomics

Author Response

Dear reviewer, my co-authors and I would like to thank you for the suggestions made during this high-quality review and then we present the answers to the questions.

We inform that with the reviews and suggestions, we were able to improve the idea presented by our work and we appreciate the opportunity. We hope this review has left the article suitable for publication in this high-impact journal and respect in the area.

Kind Regards.

Response to reviewer 2

Major Comments:

The manuscript has been poorly written. There are loads of typos and misspellings, and references are lacking in places.

R: The manuscript went through full revision of the written text and elaborated figures for orthography and formatting corrections.

The generated data by the used bioinformatic tools could be a starting point to obtain preliminary data to further study the role of miRNAs in HTLV-1 infection. However, this does not provide enough data to publish the findings in its current state.

R: Our bioinformatics analysis was used to support the general context of this literature review and it follows the concept of commonly published high-impact papers such as Karimzadeh et al. Cancer Gene Therapy 2020 (doi: 10.1038/s41417-020-0172-0). Regardless, the manuscript went through extensive revisions as section 3 was further divided into two topics so that it could be discussed in more detail.

Minor Comments:

The suggested revisions have been accepted and corrected along the text. Some that are more relevant will be commented below

“Line 96 …: HTLV-1 increases genomic instability by direct actions to the DNA…this is followed by references in Portuguese. Whilst it is great that these references are included, it would be helpful to the reader to explain in a bit more detail what was described in these papers.”

R: These references in line 96 have been changed to ones written in English. While references in Portuguese still remain in the article, they are now always accompanied by references in English referring to the same topics.

“Figure one can show the cell-to-cell transmission or transmission by mitosis, as described at line 99-105.”

R: Figure 1 is already filled with text and visual information and the authors believe it would be best to maintain the current format to avoid image-visual pollution and misunderstandings due to an excess of crowded information.

“Figure 1: retroviruses do not assemble with intact capsid cores, the cores become mature following protease dependent maturation”

R: The protease-dependent maturation has now been cited in the figure subtitle.

“Have you already defined the accessory proteins of HTLV-1?”

R: Reference 34 has been added where accessory proteins are mentioned so that the interested reader may dive deeper into their identities and characteristics

“I failed to relate the line 136-140 to the whole concept in general and to the previous paragraph particularly. What is the point of floral cells here?”

R: The appointed paragraph has been moved to a more proper location and the presence of flower cells has been further defined in text.

Round 2

Reviewer 1 Report

Finally, the manuscript has been sufficiently organized and improved. English spell check is required (e.g., TGF-B in lines 375, 410, 411, and so on, HLTV-1 in line 473).

Author Response

Dear reviewer, the minor adjustments were made. We really appreciate the comments.

Kind Regards